

# Diet breadth and exploitation of exotic plants shift the core microbiome of *Cephaloleia*, a group of tropical herbivorous beetles

Chelsea L. Blankenchip, Dana E. Michels, H. Elizabeth Braker and
Shana K. Goffredi

Department of Biology, Occidental College, Los Angeles, CA, USA

## ABSTRACT

The beetle genus *Cephaloleia* has evolved in association with tropical ginger plants and for many species their specific host plant associations are known. Here we show that the core microbiome of six closely related Costa Rican *Cephaloleia* species comprises only eight bacterial groups, including members of the *Acinetobacter*, Enterobacteriacea, *Pseudomonas, Lactococcus,* and *Comamonas*. The *Acinetobacter* and Enterobacteriacea together accounted for 35% of the total average 16S rRNA ribotypes recovered from all specimens. Further, microbiome diversity and community structure was significantly linked to beetle diet breadth, between those foraging on less than two plant types (specialists) versus over nine plant types (generalists). Moraxellaceae, Enterobacteriaceae, and Pseudomonadaceae were highly prevalent in specialist species, and also present in eggs, while Rickettsiaceae associated exclusively with generalist beetles. Bacteria isolated from *Cephaloleia* digestive systems had distinct capabilities and suggested a possible beneficial role in both digestion of plant-based compounds, including xylose, mannitol, and pectin, and possible detoxification, via lipases. *Cephaloleia* species are currently expanding their diets to include exotic invasive plants, yet it is unknown whether their microbial community plays a role in this transition. In this study, colonization of invasive plants was correlated with a dysbiosis of the microbiome, suggesting a possible relationship between gut bacteria and niche adaptation.

## INTRODUCTION

Mutually beneficial symbioses are the rule, rather than the exception, and the discovery and elucidation of the role of symbiotic microorganisms to animal life has emerged as an important area of research. Among insects, persistent bacterial partnerships are well documented and believed to play a critical role in host adaptation to specific niches. Microbiological studies on aphids, stinkbugs, psyllids, white flies, mealybugs, and leaf-hoppers have provided insights into the physiological, ecological, and evolutionary history of bacterial symbioses with insects that primarily consume plant sap (reviewed in

Corresponding author
Shana K. Goffredi,
sgoffredi@oxy.edu

*Moran & Telang, 1998*; *Wernegreen, 2002*; *Douglas, 2009*; *Bistolas et al., 2014*). It is widely accepted that symbioses allow insect herbivores to exploit plants more effectively, however little specific evidence has been reported for chewing phytophagous insects, particularly in the understudied tropical rainforests (*Genta et al., 2006*; *Kuriwada et al., 2010*; *Hansen & Moran, 2014*).

With more than 200 described species, the neotropical beetle genus *Cephaloleia* (Chevrolat) has evolved in specific association with gingers in the order Zingiberales (*Staines, 1996*; *Wilf et al., 2000*; *McKenna & Farrell, 2006*; *García-Robledo & Staines, 2008*). For ∼50 MY, *Cephaloleia* beetles have specialized on the immature rolled leaves of these gingers, which they use for nutrition, development, reproduction, and as shelter (*McKenna & Farrell, 2005*). Unlike most tropical insects, host plant associations are known for most of the sympatric species of *Cephaloleia* that inhabit the lowland rainforest in and around La Selva Biological Station, Costa Rica (*Staines, 2011*; *García-Robledo & Staines, 2014*). As a group, they display variability in diet breadth, ranging from hypergeneralist species, such as *Cephaloleia belti* (Baly) which feeds on 15+ plants from three Zingiberales families, to specialist species, such as *C. placida* (Baly) which is only found on a single plant species (*McKenna & Farrell, 2005*; *García-Robledo et al., 2013*). Studies exploring the link between gut bacterial community and diet in non-sap sucking insects have produced conflicting results (*Colman, Toolson & Takacs-Vesbach, 2012*; *Jones, Sanchez & Fierer, 2013*; *Rahman et al., 2015*), and very few have examined closely related insects with both generalist and specialist feeding strategies. Fierer and colleagues recommended that in order to resolve the effects of diet on the microbiome, future studies should focus on a single group of insects with varied diets (*Jones, Sanchez & Fierer, 2013*). Thus, the contrasting dietary breadths of *Cephaloleia* beetles make them an ideal natural group by which to examine the, perhaps reciprocal, role of the microbiome on niche adaptation. Additionally, in the last two decades, at least four Zingiberales from South America have invaded the tropical rainforest of La Selva, including the pink velvet banana and white ginger lily (*García-Robledo & Horvitz, 2011*). Interestingly, at least eight *Cephaloleia* beetle species, including several specialists, are currently expanding their diets to exploit these novel Zingiberales (*García-Robledo & Horvitz, 2011*).

Understanding the presence, diversity, and pervasiveness of bacteria associated with insects, especially in relation to diet breadth, is a necessary and integral component of insect nutritional ecology (*Douglas, 2013*). In this paper, we sought to characterize the diversity of bacteria, via 16S rRNA sequencing and bacterial cultivation, associated with adults (and a small number of eggs) of six species of *Cephaloleia* beetles, including two generalist and four specialist species. Initially we hypothesized that generalist species would have a more diverse microbiome, as an adaptation to a wide range of plant types or as a consequence of increased encounters with diverse bacteria associated with plant tissues (*Hansen & Moran, 2014*; *Engel & Moran, 2013*). Major beetle bacterial groups were cultured to determine their metabolic capabilities and whether they might aid in plant digestion or detoxification of plant compounds by the host insect. Three *Cephaloleia* species were collected from invasive white ginger and pink banana, including the

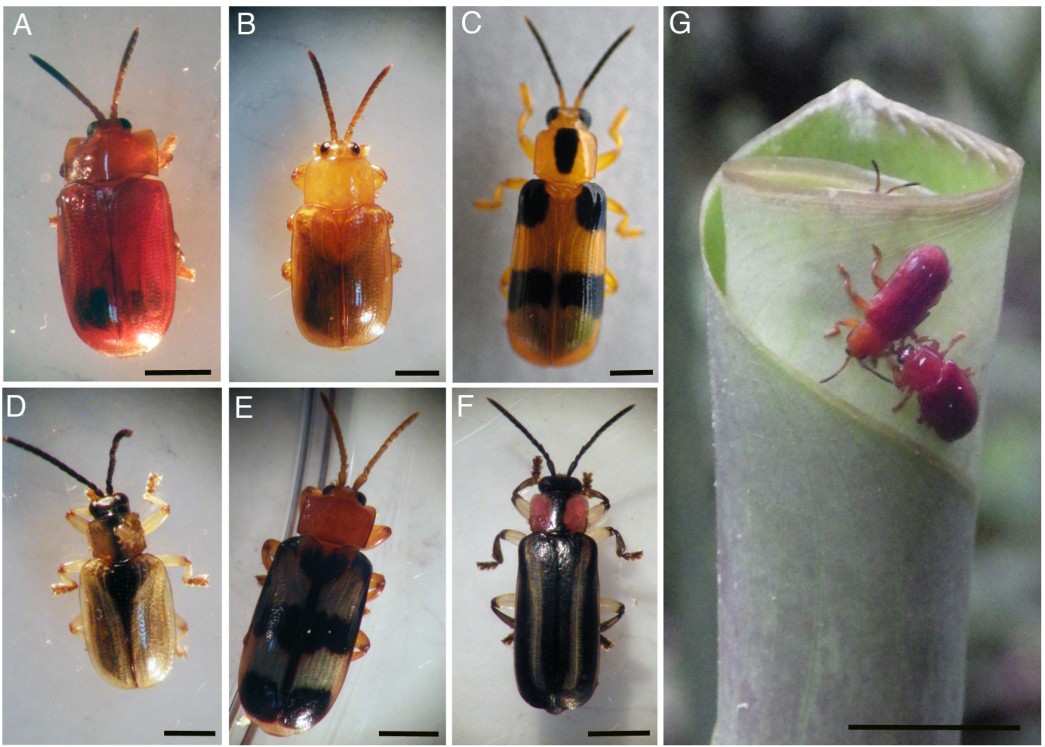

**Figure 1 Costa Rican *Cephaloleia* species in this study.** Costa Rican *Cephaloleia* species in this study include (A) *C. dilaticollis*, (B) *C. placida*, (C) *C. reventazonica*, (D) *C. dorsalis*, (E) *C. fenestrata*, and (F) *C. belti*. (G) A mating pair of *C. erichsonii* on a young rolled leaf of *Calathea lutea*. All life stages and most behavior, including mating, take place on plants within the order Zingiberales. Scale bars = 1 mm for all images, except far right where the scale bar is 1 cm. Photo credits: S. Goffredi.

generalist *C. belti* and specialists *C. placida* and *C. dilaticollis* (Baly). Comparisons between beetles feeding on native versus invasive plants was expected to reveal specific bacterial groups related to colonization of exotic plants and provide a better understanding of the role of the microbiome in adaptation.

## MATERIALS AND METHODS

### Specimen collection

Adult beetles ($n = 38$; Fig. 1) were collected in 2014–2016 at La Selva Biological Station, a 1,500 ha ecological reserve in a lowland tropical rainforest site in northern Costa Rica (10°26′N, 83°59′W; current Costa Rican Ministry of the Environment and Energy permit #R-026-2015-OT-CONAGEBIO). Adults were located by searching for rolled Zingiberales leaves that were then unrolled and beetles collected with forceps. Beetles with excised exoskeletons were preserved in 70% ethanol, at 4 °C, for later molecular analysis. *Cephaloleia* eggs ($n = 5$) were collected adhered to plastic after mating pairs were kept for a brief time in captivity in bags. Usually the adults laid eggs on the plastic, which was remote from where they spent most of their time (in a rolled leaf near the bottom of the bag, or in the zipper seals of the bag). The piece of plastic with attached egg was then surface sterilized and kept in ethanol until analysis. Any bacterial presence in the eggs

was therefore attributed to vertical transmission by adults, and not due to adult contamination of the eggs. For bacterial cultivation, the digestive systems of the beetles were dissected in sterile $1\times$ phosphate buffered saline (PBS) and homogenized using a ground glass mortar and pestle. The resulting homogenate was spread onto Trypticase Soy Agar (TSA) and incubated at ambient temperatures. Resulting bacterial colonies were selected and stored at $-80\ °C$ in 30% glycerol (in $1\times$ PBS). All samples were transported back to the US for further processing.

## Specimen identification

Adult beetles were identified based on diagnostic morphology and host plant (the latter mainly for specialist beetles that feed on only a single plant type; *McKenna & Farrell, 2005*; *García-Robledo et al., 2013*). In one cased we employed DNA analysis for identification; *C. dilaticollis* currently encompasses two cryptic species, one of which is a specialist and the other a generalist. Total genomic DNA was extracted using the Qiagen DNeasy Kit (Qiagen, Valencia, CA, USA) according to the manufacturer's instructions. The cytochrome c oxidase I (COI) gene was amplified via the polymerase chain reaction (PCR) using the insect COI primers 1718F (5′-GGAGGATTTGGAAATTGATTAGTTCC-3′) and 3661R (5′-CCACAAATTTCTGAACATTGACCA-3′) according to *McKenna & Farrell (2005)*. Successful PCR reactions, determined via electrophoresis, were cleaned using MultiScreen HTS plates (Millipore Corporation, Bedford, MA, USA) and sequenced via Laragen, Inc. (Los Angeles, CA, USA). Beetle species identification was confirmed based on COI sequences, upon consultation with Dr. Carlos García-Robledo (University of Connecticut).

## Molecular microbiome analysis

Adult *Cephaloleia* beetles from six species found on native plants ($n = 29$) and three species found on invasive plants ($n = 9$) were examined for microbiome composition via 16S rRNA gene barcode sequencing (Table 1). As described above, total genomic DNA was extracted using the Qiagen DNeasy Kit (Qiagen, Valencia, CA, USA) according the manufacturer's instructions. All extractions were from individual beetles, with the exception of *C. dorsalis* (Baly) which due to its small size, two to three individuals were pooled to achieve positive PCR results. PCR amplification of the bacterial 16S rRNA gene was performed using the specific primers, 515F (5′-GTGCCAG-CMGCCGCGGTAA-3′) and 806R (5′-GGACTACHVGGGTWTCTAAT-3′; *Caporaso et al., 2011*). The thermal cycling profile used was as follows: an initial denaturation at 94 °C, then 45 s at 94 °C, 1 min at 50 °C, and 90 s at 72 °C, for 29 cycles, followed by 10 min at 72 °C. Successful PCR amplifications, assessed via electrophoresis, were pooled, in duplicate, and barcodes were added according to the Earth Microbiome Project (EMP; *Caporaso et al., 2011*); 5 μl of the amplicon product from PCR#1 was used as template in a five cycle, 25 μl reconditioning reaction with the same EMP-recommended conditions and the full EMP primers (515f_barcode: AATGATACGGCGAC-CACCGAGATC-TACACTATGGTAATTGTGTGCCAGCMGCCGCGGTAA; 806r_barcode: CAAGCAGAA-GACGGCATACGAGAT-X-AGTCAGTCAGCCGGACTACHVGGGTWTCTAAT), where X indicates a unique 12 bp barcode. Adding the barcode indices at the second step

**Table 1 Beetle specimens analyzed in this study.**

| Species | Plant Diet[a] | Sample ID | 16S rRNA sequences (initial #) | 16S rRNA sequences (normalized)[b] | Shannon diversity value[c] |
|---|---|---|---|---|---|
| **Native plants** | | | | | |
| C. belti | H. latisplatha | Cb1 | 35380 | 12958 | 2.50 |
| | H. latisplatha | Cb2 | 41041 | 13039 | 2.29 |
| | H. latisplatha | Cb3 | 40375 | 12939 | 2.14 |
| | H. wagneriana | Cb4 | 35682 | 13276 | 1.80 |
| | Calathea sp. | Cb5 | 23717 | 13158 | 1.73 |
| | H. latisplatha | Cb6 | 21909 | 13186 | 0.82 |
| | unknown | Cb7 | 43649 | 13018 | 1.94 |
| | H. latisplatha | Cb8 | 20413 | 12959 | 2.26 |
| C. reventazonica | H. imbracata | Cr1 | 20451 | 13177 | 1.89 |
| | H. imbracata | Cr2 | 42274 | 13237 | 1.16 |
| | H. imbracata | Cr3 | 32029 | 13324 | 1.81 |
| C. fenestrata | Calathea sp. | Cf1 | 67826 | 12572 | 3.00 |
| | Calathea sp. | Cf2 | 49366 | 12887 | 3.12 |
| | Calathea sp. | Cf3 | 15568 | 13151 | 2.21 |
| | Calathea sp. | Cf4 | 26983 | 12894 | 2.29 |
| | Calathea sp. | Cf5 | 32891 | 12758 | 2.46 |
| | H. imbracata | Cf6 | 31801 | 12637 | 2.11 |
| | H. imbracata | Cf7 | 48964 | 12969 | 3.29 |
| C. dorsalis | Co. malortiensus | Cdor1 | 41144 | 12755 | 2.97 |
| | Co. malortiensus | Cdor2 | 54265 | 9899 | 2.91 |
| | Co. malortiensus | Cdor3 | 51930 | 12711 | 2.05 |
| C. dilaticollis | R. alpinia | Cdil1 | 45418 | 13084 | 2.38 |
| | R. alpinia | Cdil2 | 47713 | 13096 | 2.47 |
| | unknown | Cdil3 | 46765 | 12948 | 2.92 |
| C. placida | R. alpinia | Cp1 | 40199 | 12979 | 2.79 |
| | R. alpinia | Cp2 | 45794 | 11664 | 1.49 |
| | R. alpinia | Cp3 | 41131 | 12782 | 2.16 |
| | R. alpinia | Cp4 | 48511 | 13158 | 2.52 |
| | R. alpinia | Cp5 | 37048 | 12770 | 3.08 |
| **Invasive plants** | | | | | |
| C. belti | M. velutina | Cb_inv1 | 49334 | 13005 | 2.08 |
| | M. velutina | Cb_inv2 | 33427 | 13238 | 1.66 |
| | M. velutina | Cb_inv3 | 26965 | 12479 | 1.97 |
| | M. velutina | Cb_inv4 | 30173 | 12983 | 2.19 |
| C. dilaticollis | He. coronarium | Cdil_inv1 | 15895 | 12268 | 2.32 |
| | H. coronarium | Cdil_inv2 | 15679 | 12650 | 2.27 |
| | H. coronarium | Cdil_inv3 | 23419 | 12473 | 3.02 |
| C. placida | H. coronarium | Cp_inv1 | 13829 | 13128 | 2.39 |
| | H. coronarium | Cp_inv2 | 26564 | 12733 | 3.15 |

(Continued)

| Species | Plant Diet[a] | Sample ID | 16S rRNA sequences (initial #) | 16S rRNA sequences (normalized)[b] | Shannon diversity value[c] |
|---|---|---|---|---|---|
| *Eggs* | | | | | |
| *C. belti* | *n/a* | Cb_egg1 | 68837 | 4410 | 2.90 |
| | *n/a* | Cb_egg2 | 55636 | 9715 | 3.36 |
| *C. dorsalis* | *n/a* | Cdor_egg1 | 45536 | 13217 | 2.84 |
| *C. dilaticollis* | *n/a* | Cdil_egg1 | 35848 | 6931 | 2.88 |
| | *n/a* | Cdil_egg2 | 18648 | 7305 | 3.55 |

Notes:
Specimens analyzed in this study, showing the # of 16S rRNA sequences generated from barcoding, along with corresponding measures of diversity.
[a] *H. = Heliconia, He. = Hedychium, M. = Musa, R. = Renealmia, Co. = Costas.*
[b] Define "normalized"—without mitochondria and chloroplasts; with *Wolbachia*.
[c] Diversity values without *Wolbachia*.

minimizes PCR bias that would result from employing long primers over many cycles (*Berry et al., 2011*). Further, the use of the "reconditioning" PCR for barcoding, as well as the pooling of duplicate amplifications ahead of barcoding, was an attempt to minimize PCR errors and bias, respectively (*Kennedy et al., 2014*). Samples were mixed together in equimolar amounts and purified in bulk through a Qiagen PCR Purification kit. At all PCR steps, amplification success and purity was checked by gel electrophoresis. Paired-end sequences (2*x* 250 bp) were generated from barcoded amplicon products at Laragen, Inc., on an Illumina MiSeq platform. At Laragen, the raw data was passed through a filter which demultiplexed the library into individual samples and removed any sequences which had >1 bp mismatch on the 12 bp barcode sequence, and assigned quality scores to each basepair call on every sequence. At the same time, adapter, barcode, and primer sequences were removed.

Sequence processing was performed in QIIME 1.8.0 (Quantitative Insights Into Microbial Ecology; *Caporaso et al., 2010*). Sequences were clustered at 99% similarity and a representative sequence from each cluster was assigned a taxonomic identification using parameter -m in QIIME and the Silva115 database. Via barcode amplicon sequencing, 13829–68837 sequences were recovered from each specimen (Table 1; Table S1). To avoid artifacts of sequencing depth, the number of sequence reads was standardized to 13829 sequences per specimen, based on the lowest sequence number for specimen "Cp_inv1" (*C. placida* on invasive white ginger; Table 1). The dataset was further cut off at 1% (i.e., the number of sequence hits for a single bacterial OTU across all 38 specimens must have been greater than 138 to be included). After this cutoff, sequences ranged from 9899 to 13324 per specimen (Table 1). *Wolbachia* was observed in 18 specimens, among three species (range of 0.1–56.6% for all sequences; *C. belti* avg 22% ± 18%, *C. reventazonica* avg 41% ± 17%, *C. fenestrata* avg 10% ± 13%,), but was removed from subsequent analysis based on its known prevalence in insects as a reproductive pathogen (Table S2). Sequences corresponding to chloroplasts and mitochondria were also removed from the data set. NMDS, ANOSIM, and SIMPER analyses were completed in Primer-E after square-root transforming the dataset and calculating Bray–Curtis similarities (*Clarke & Warwick, 2001*). An ANOSIM R value close to "1.0" suggests

dissimilarity between groups. Close environmental and cultured relatives were chosen using top hits based on BLAST (www.ncbi.nlm.nih.gov). RStudio was used to perform ANOVA calculations using a script available at https://sites.oxy.edu/sgoffredi/Symbiosis_Lab/LabScripts.html.

### *Cephaloleia belti* phenotypes and diagnostic PCR

*Cephaloleia belti* individuals ($n = 45$) were photographed and sized (length and width at pronotum) using imageJ (*Schneider, Rasband & Eliceiri, 2012*). The eight largest and eight smallest beetles were dissected for molecular analysis according to the methods described above. Total DNA of the body of the beetles was extracted using the Qiagen DNeasy Kit (Qiagen, Valencia, CA, USA) according the manufacturer's instructions. For these 16 beetles, a diagnostic PCR using two different sets of pathogen-specific PCR primers was performed specifically for the bacterial genera *Rickettsia* (Rsp-F 5′-CGCAACCCTCATTCTTATTTGC-3′, Rsp-R 5′-CCTCTGTAAACACCAT-TGTAGCA-3′; *Giulieri et al., 2012*) and *Spiroplasma* (Spiro_16SF 5′-GGTCTTCGGATTGTAAAGGTCTG-3′, Spiro_16SR 5′-GGTGTGTACAAGACCCGAGAA-3′; *Haselkorn, Markow & Moran, 2009*) with the following thermal protocol: an initial 5 min denaturation at 94 °C, then 1 min at 94 °C, 1 min at 56 °C, and 1 min at 72 °C, for 29 cycles, and a final 5 min extension at 72 °C. Successful PCR amplification was determined via electrophoresis and confirmed to be *Rickettsia* or *Spiroplasma* via direct Sanger sequencing (Laragen, Inc., Los Angeles, CA, USA).

### Bacterial cultivation

Initial bacterial suspensions in 30% glycerol stocks were re-grown on TSA plates at 30 °C. Growth was checked for morphological purity before being suspended in 40 μl of alkaline PEG (60 g of PEG 200 with 0.93 ml of 2 M KOH and 39 ml of water). This suspension was then heated to 96 °C for 20 min in order to lyse the bacterial cells and liberate the DNA. The 16S rRNA gene was then amplified directly using the general PCR primers 27F and 1492R (*Lane, 1991*) and the following thermal protocol: an initial 5 min denaturation at 94 °C, followed by 94 °C for 45 s, 54 °C for 1 min, and 72 °C for 90 s, for 29 cycles, and a final 72 °C extension for 10 min. Successful amplifications were checked via electrophoresis, cleaned using MultiScreen HTS plates (Millipore Corporation, Bedford, MA, USA), and sequenced at Laragen, Inc. Sequences were compared with the NCBI BLAST database to determine bacterial identity. Bacteria were propagated on TSA plates to ensure proper activity prior to metabolic testing. The ability to digest lactose/glucose, xylose, mannitol, and pectin was determined using phenol red agar (HiMedia, with 10% of each substrate). Protein digestion was determined using Litmus Milk tubes purchased from Carolina Biological Supply Company (Burlington, NC, USA). The ability to breakdown lipids was tested using an APIZYM analysis (bioMerieux, Inc., Durham, NC, USA), according to the manufacturer's instructions.

### Data availability

The raw barcode sequence data are available from the Dryad Digital Repository: http://dx.doi.org/10.5061/dryad.5fj6t. Raw sequences were aligned and quality control for

unidentified base pairs and chimeras was performed according to the specifics noted at
https://sites.oxy.edu/sgoffredi/Symbiosis_Lab/LabScripts.html. The QIIME processed
data are also available from the Dryad Digital Repository: http://dx.doi.org/10.5061/
dryad.5fj6t. 16S rRNA sequences for bacterial isolates are available from GenBank under
accession numbers MF776885–MF776899.

# RESULTS

## The limited core microbiome of *Cephaloleia* beetles

Using barcode 16S rRNA analysis, the microbiome of adults of six species of *Cephaloleia*,
foraging on native plant diets, was characterized taxonomically (Table 1). Collectively,
168 bacterial OTU's ("operational taxonomic unit;" defined as 99% sequence similarity)
were recovered from all adult specimens examined ($n = 29$), while individual beetles
generally associated with 47–152 OTUs ($100 \pm 31$). Greater than 60% of *Cephaloleia*
specimens contained a core group of eight bacterial OTUs, including three members of
the *Acinetobacter*, two undefined Enterobacteriacea, *Pseudomonas*, *Lactococcus*, and a
*Comamonas* (Figs. 2 and 3; Table S1). These eight bacterial OTUs comprised the majority
of 16S rRNA sequences recovered from each *Cephaloleia* individual (up to 88%).

A single bacterial family, Moraxellaceae, dominated the microbiomes of all 29
specimens feeding on native plants, combined (representing 24% of the total recovered
sequences). The genus *Acinetobacter*, in particular, comprised the vast majority of the
Moraxellaceae sequences and accounted for 23% of the 16S rRNA sequences recovered
overall. Of the 19 different *Acinetobacter* OTUs, three were responsible for 60% of the
total *Acinetobacter* diversity and were each present in >75% of beetles (OTUs-131476,
21817, and 28305; Fig. 3, shown in purple; Table S1), suggesting them to be members
of the core *Cephaloleia* microbiome. *Acinetobacter* OTU131476 was found in 22 of
29 specimens, and was 11% abundant (on average, for all sequences recovered in each of
29 beetles found on native plants; Fig. 3). This OTU was 98% similar to bacteria associated
with both leaf cutter ants and fig wasps (GenBank accession #'s LN564930, HQ639556).
*Acinetobacter* OTU21817 was also found in 22 of 29 specimens, represented 6% average
abundance (Fig. 3), and was 100% similar to bacteria found in the midgut of a
leafworm moth (GenBank accession # KU841476). *Acinetobacter* OTU28305 was found
within 23 of 29 specimens, totaling 6% average abundance (Fig. 3), and was 100%
identical to *Acinetobacter baylyi* (GenBank accession # NR115042), and others found in
the rhizosphere.

Unidentified Enterobacteriaceae were also dominant in beetles found on native plants,
representing 12% of the total recovered sequences. Of the 10 OTUs that comprised the
Enterobacteriaceae within *Cephaloleia* beetles, a single OTU was responsible for 65% of
the total Enterobacteriaceae diversity. This dominant Enterobacteriaceae OTU-79811 was
present in >93% of beetle specimens, was 8% abundant on average (for all 29 beetles
found on native plants; Fig. 3, shown in blue), and was 100% similar to *Enterobacter/
Klebsiella* bacteria recovered from scarab beetles, sand flies, and pill bugs. Two additional
OTUs (OTU127346 and OTU79806) each accounted for ~15% of the remaining
Enterobacteriaceae and were present in >15 of 29 specimens (Fig. 3). These OTUs were

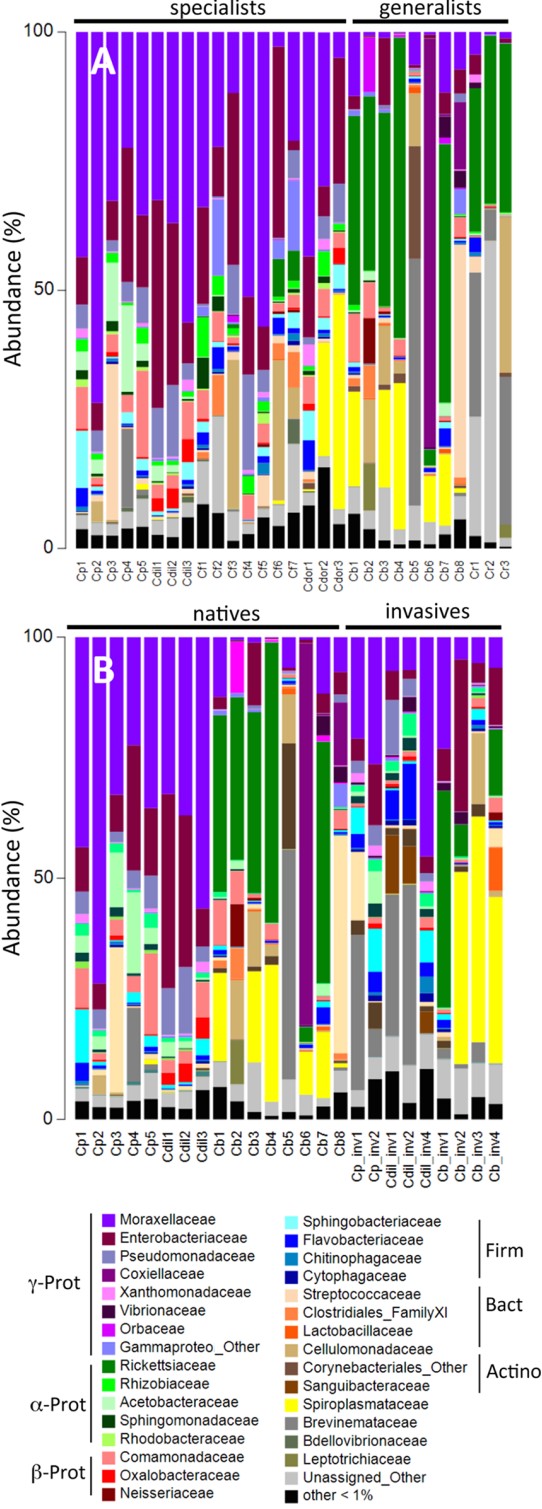

**Figure 2 Relative abundance of bacterial families.** Relative abundance of bacterial families from (A) beetles foraging on native plants, including *Cephaloleia placida*, *C. dilaticollis*, *C. fenestrata*, *C. doralis*, *C. belti*, and *C. reventazonica*, and (B) beetles foraging on invasive plants *C. placida*, *C. dilaticollis*, and *C. belti* compared to those species foraging on native plants. Each color group on the graph represents a distinct genus-level OTU or lowest level available. Families that constituted <1% of sequences from individual specimens were grouped as "Other."

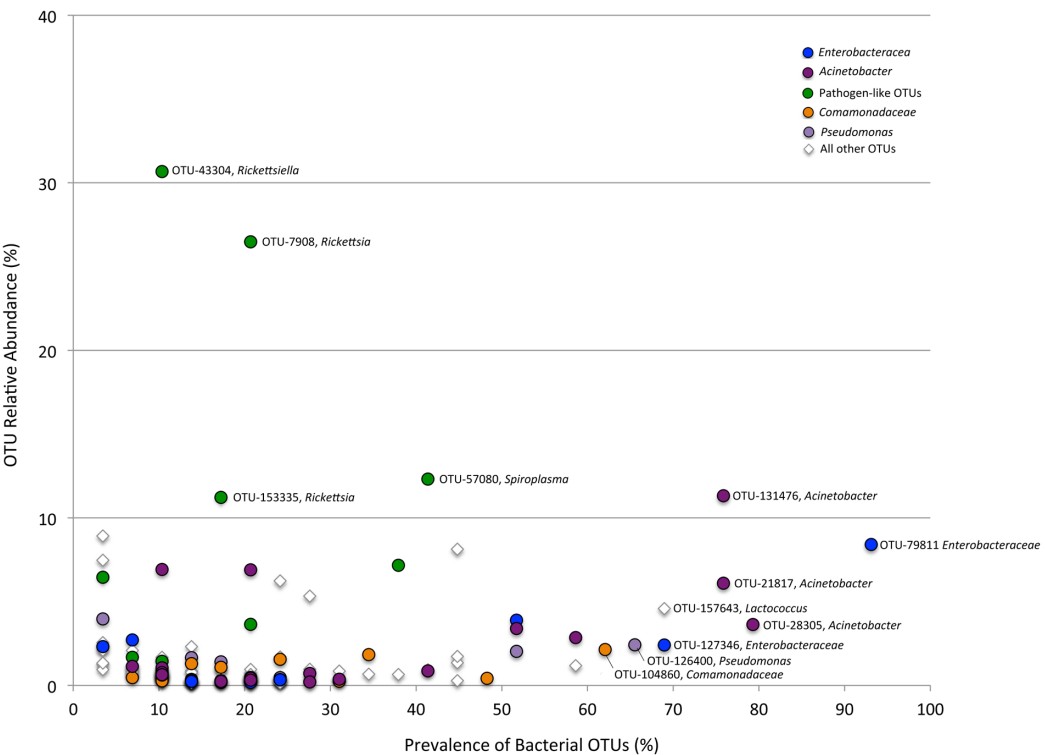

**Figure 3 Prevalence versus relative abundance of bacterial OTUs.** Prevalence versus relative abundance of bacterial OTUs associated with six *Cephaloleia* species (*n* = 29 specimens, collected on native diets). Certain dominant OTU groups are indicated separately by color (e.g., *Acinetobacter*), with 12 shown by OTU# based on ≥60% prevalence or ≥10% relative abundance (including four noted in green that matched bacterial groups typically thought of as pathogens).

related to *Citrobacter* and *Raoultella* OTUs found in the microbiome of numerous insects, including honeybees (KR269812), scarab beetles (KT956239), and fruit flies (KX997073).

Three additional OTUs were highly prevalent in *Cephaloleia* microbiomes (present in ~64% of individual beetles), including a *Lactococcus* OTU-157643 representing an average abundance of ~4% (Fig. 3), a *Pseudomonas* OTU-126400 with an average abundance of ~2% (related to bacteria recovered from mosquitoes and sand flies; KY041526; *Li et al., 2016*), and a *Comamonas* OTU-104860, also with an average abundance of ~2% (most closely related to bacteria found in association with fruit flies; KX994588; Fig. 3; Table S1).

A large number of cultured isolates recovered from the digestive systems of *Cephaloleia* (81% of 37 isolated bacterial colonies) were members of the Moraxellaceae, Enterobacteriaceae, and Pseudomonadaceae, based on 16S rRNA gene sequencing. Several isolates had 16S rRNA sequences identical to the dominant bacteria identified via barcode 16S rRNA sequencing, including Acineto3 (=*Acinetobacter* OTU-28305), Entero4 (=Enterobacteriaceae OTU-127346), and Pseudo2 (=*Pseudomonas* OTU-126400). The Enterobacteriaceae were found to utilize plant-based compounds, including xylose and pectin (seven of nine isolates), mannitol (eight isolates), and lactose/glucose (all nine

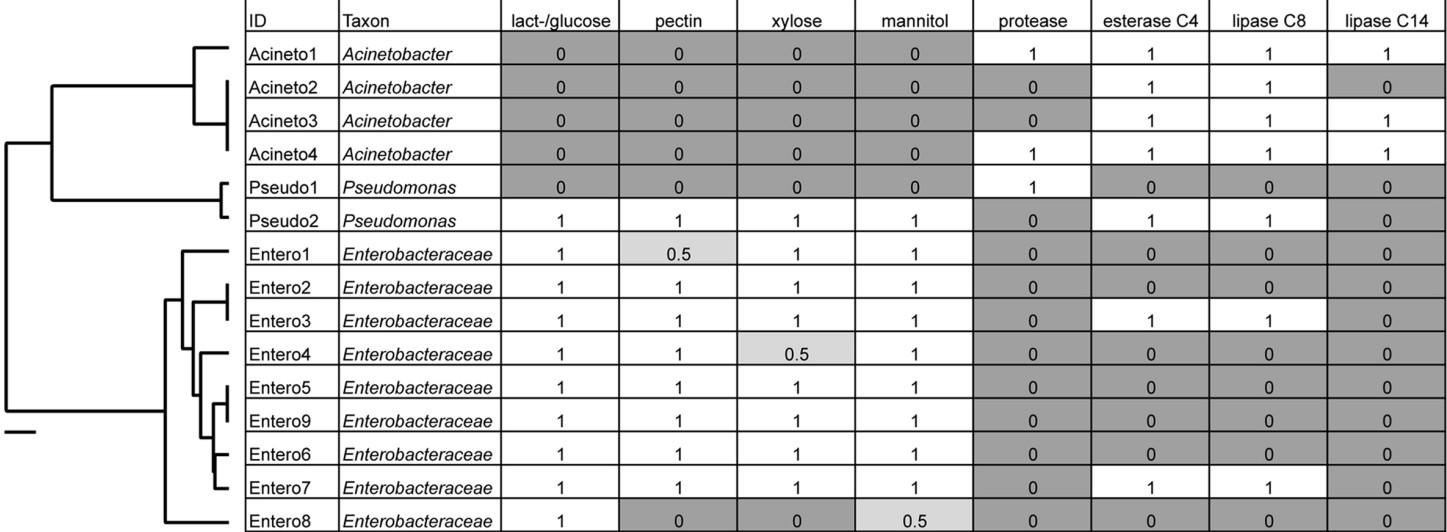

| ID | Taxon | lact-/glucose | pectin | xylose | mannitol | protease | esterase C4 | lipase C8 | lipase C14 |
|---|---|---|---|---|---|---|---|---|---|
| Acineto1 | *Acinetobacter* | 0 | 0 | 0 | 0 | 1 | 1 | 1 | 1 |
| Acineto2 | *Acinetobacter* | 0 | 0 | 0 | 0 | 0 | 1 | 1 | 0 |
| Acineto3 | *Acinetobacter* | 0 | 0 | 0 | 0 | 0 | 1 | 1 | 1 |
| Acineto4 | *Acinetobacter* | 0 | 0 | 0 | 0 | 1 | 1 | 1 | 1 |
| Pseudo1 | *Pseudomonas* | 0 | 0 | 0 | 0 | 1 | 0 | 0 | 0 |
| Pseudo2 | *Pseudomonas* | 1 | 1 | 1 | 1 | 0 | 1 | 1 | 0 |
| Entero1 | *Enterobacteraceae* | 1 | 0.5 | 1 | 1 | 0 | 0 | 0 | 0 |
| Entero2 | *Enterobacteraceae* | 1 | 1 | 1 | 1 | 0 | 0 | 0 | 0 |
| Entero3 | *Enterobacteraceae* | 1 | 1 | 1 | 1 | 0 | 1 | 1 | 0 |
| Entero4 | *Enterobacteraceae* | 1 | 1 | 0.5 | 1 | 0 | 0 | 0 | 0 |
| Entero5 | *Enterobacteraceae* | 1 | 1 | 1 | 1 | 0 | 0 | 0 | 0 |
| Entero9 | *Enterobacteraceae* | 1 | 1 | 1 | 1 | 0 | 0 | 0 | 0 |
| Entero6 | *Enterobacteraceae* | 1 | 1 | 1 | 1 | 0 | 0 | 0 | 0 |
| Entero7 | *Enterobacteraceae* | 1 | 1 | 1 | 1 | 0 | 1 | 1 | 0 |
| Entero8 | *Enterobacteraceae* | 1 | 0 | 0 | 0.5 | 0 | 0 | 0 | 0 |

**Figure 4 Metabolic capabilities of bacteria isolated from the digestive system of *Cephaloleia* beetles.** Metabolic capabilities of bacteria isolated from the digestive system of *Cephaloleia* beetles, including the ability to use lactose/glucose, pectin, xylose, and mannitol, as well as the presence of proteases, esterases, and lipases. White shading and the number "1" indicate the ability to digest the specified compound. Dark gray shading and the number "0" indicate an inability to digest the specified compound, "0.5" indicates a partial ability. At left, a phylogenetic tree, based on 767 bp 16S rRNA sequences, built with Tamura–Nei distance model and UPGMA method, of beetle digestive system isolates shown to the left. Scale bar, 0.1 divergence.

isolates). In contrast, none of the four *Acinetobacter* isolates in this study were able to digest these compounds, but instead uniquely displayed esterase C4, lipase C8, and lipase C14 capabilities (Fig. 4).

## Diet breadth influences the microbiome of *Cephaloleia* beetles

Overall, the microbiome of specialist beetle species was significantly higher in diversity than generalists (2.6 ± 0.5 versus 1.9 ± 0.5, respectively; $p = 0.0006$, one-way ANOVA, Fig. 5). Measures of bacterial diversity (via the Shannon diversity index) were 0.8–2.5 for *C. belti*, 1.1–1.9 for *C. reventazonica*, 2.1–3.3 *C. fenestrata*, 2.0–3.0 for *C. dorsalis*, 2.4–2.9 for *C. dilaticollis*, and 1.5–3.0 for *C. placida* (Table 1). Further, NMDS ordination revealed the microbial assemblages of *Cephaloleia* to be strongly differentiated by diet breadth (i.e., generalist versus specialist; $R = 0.74$, $p = 0.001$, analysis of similarity (ANOSIM); Fig. 6A). SIMPER analysis implicated several bacterial families associated with this difference. For example, the Moraxellaceae, Enterobacteriaceae, and Pseudomonadaceae comprised a significantly higher percentage of the bacterial community in beetle species categorized as specialists (34%, 17%, and 6% of recovered sequences on average for specialist individuals, respectively) versus generalists (4%, 3%, 0.3%, respectively; all $p < 0.0013$, one-way ANOVA; Figs. 2 and 5). In contrast, results indicate that generalist *Cephaloleia* beetles were colonized by bacteria traditionally thought of as pathogens, including *Rickettsia* and *Rickettsiella* (discussed in more detail in supplemental information results). The Rickettsiaceae comprised a significantly higher percentage of the bacterial community in beetle species categorized as generalists, *C. belti* and *C. reventazonica* (28% of recovered sequences on average) versus specialists

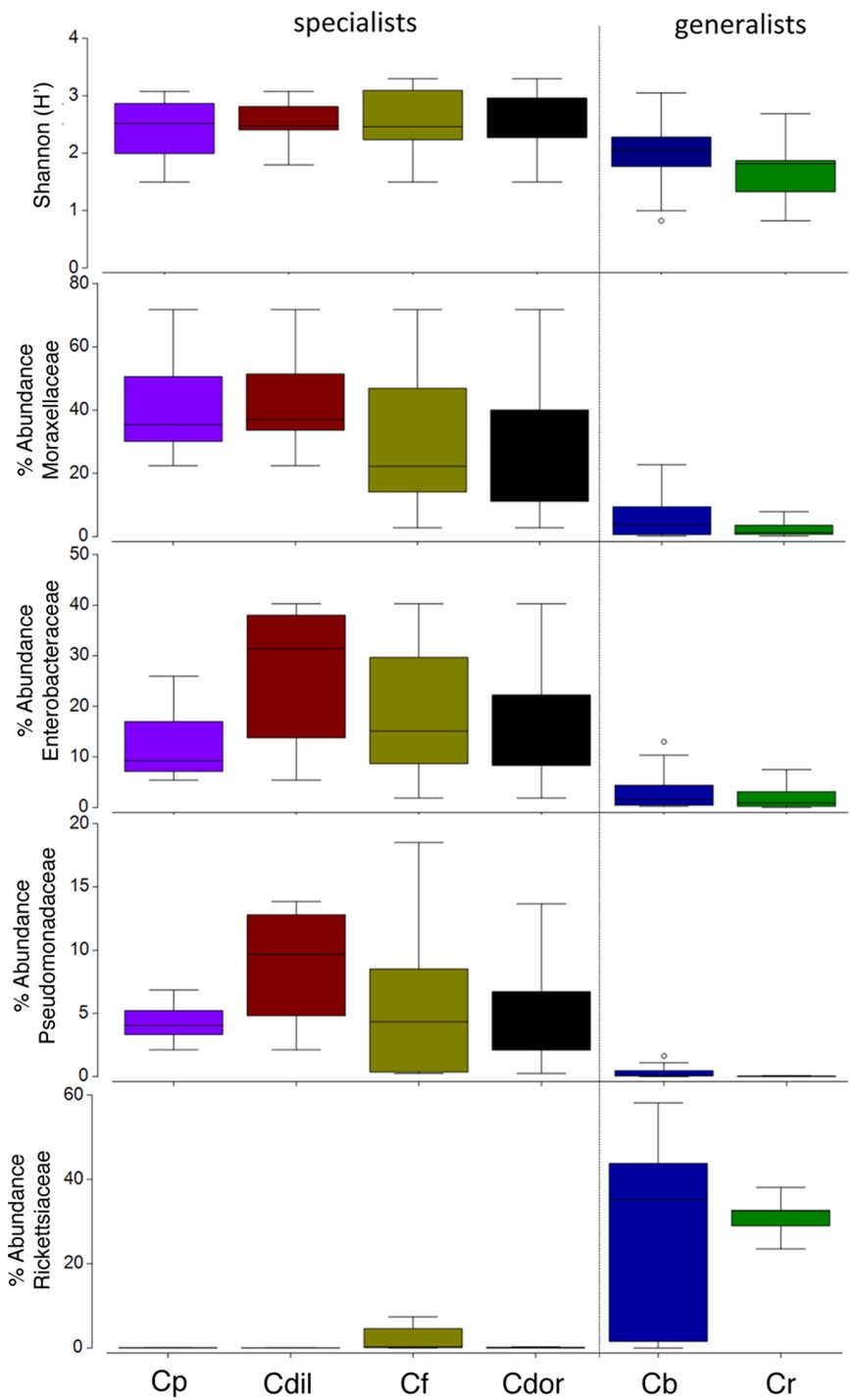

**Figure 5 Box plots of beetle-associated average diversity (Shannon index) and relative percent abundance of four key bacterial families.** Box plots of beetle-associated average diversity (Shannon index) and relative percent abundance of four key bacterial families (identified by SIMPER to be responsible for up to 46% of the cumulative (dis)similarity among six *Cephaloleia* species examined in this study ($n$ = 29 specimens, collected on native diets)). Any data points outside of the 25–75% range are identified by open symbols. Species abbreviations are as follows: plac = *C. placida*, dilat = *C. dilaticollis*, fen = *C. fenestrata*, dor = *C. dorsalis*, belti = *C. belti*, rev = *C. reventazonica*.

(only 0.8%; $p < 0.0001$, one-way ANOVA; Fig. 4; Table S1). For example, two *Rickettsia* OTUs (OTU7980 and OTU153335; Rickettsiaceae) were collectively present in only 11 out of 29 specimens, but comprised 11–26% abundance on average (for all 29 beetles found on native plants; Fig. 3). These OTU's were 99% similar to *Rickettsia* found in leafhoppers (KR709154) and ticks (MF002591), to name a few. Similarly, a *Spiroplasma* OTU (Spiroplasmataceae) was present in 12 of 29 specimens, represented an average abundance of ~12%, and was primarily observed in four individuals, two *C. dorsalis* and two *C. belti* (Figs. 2 and 3). This OTU57080 was 100% similar to the *Spiroplasma* associated with *Drosophila*. The incidence of *Rickettsia* and *Spiroplasma* was determined, via diagnostic PCR, to be highest in the smallest individuals of *C. belti* (81% prevalence, $n = 8$) versus the largest (56% prevalence, $n = 8$). Finally, a single OTU of *Rickettsiella* (Coxielliaceae) was only found in three *C. belti* individuals, but was highly abundant (~31% on average; Fig. 3). This OTU43304 was 99% similar to *Rickettsiella* bacteria, also found in sand flies and ticks.

## Bacteria associated with the eggs of *Cephaloleia* beetles found on native plants

Similar to the adults, *Cephaloleia* eggs were examined for microbiome composition via 16S rRNA gene barcode sequencing (Table 1). Four of the most common bacterial OTUs associated with adult beetles (*Acinetobacter*, Enterobacteriaceae, *Comamonas*, and *Pseudomonas*) were consistently observed in eggs (Fig. 7; Table 1), suggesting likely pseudovertical transmission from mother to offspring. NMDS analysis revealed a general overlap of the microbial communities associated with eggs and adults (ANOSIM $R = 0.19$, $p = 0.060$; Fig. 6B). The egg-associated microbiome of the generalist *C. belti* was significantly different from the adults (ANOSIM $R = 0.95$, $p = 0.022$), based mainly on a near absence of *Rickettsia* in the eggs (only 0.01% abundance; Fig. 7; Table 1).

## *Cephaloleia* beetles foraging on invasive plants have distinct microbiomes

The specialist *Cephaloleia* species collected on invasive plants exhibited an apparent dysbiosis in their microbiome. NMDS ordination revealed a distinct bacterial community structure between the specialist beetles collected from native plants, compared to those on invasive plants, including *C. placida* and *C. dilaticollis* both on white ginger (*Hedychium coronarium*; ANOSIM $R = 0.97$, $p = 0.001$; Fig. 6C). The specialist species found on invasive plants possessed a lower abundance of both Moraxellaceae (21% average 16S rRNA abundance when on invasive plants compared to 41% for native feeders; $p = 0.0472$, one-way ANOVA) and Enterobacteriaceae (6% average abundance in beetles feeding on invasive plants, as opposed to 18% for those feeding on native plants; $p = 0.0357$; Fig. 2; Table S1). This decrease in typical microbiome membership may relate directly to a concomitant increase in microbiome members such as Brevinemataceae, which was significantly more abundant in both specialist species on invasive plants (21% average abundance; a single OTU43892, ~99% similar to *Brevinema* found in insect larvae and other invertebrates, represented 97% of all Spirochaete sequences), compared

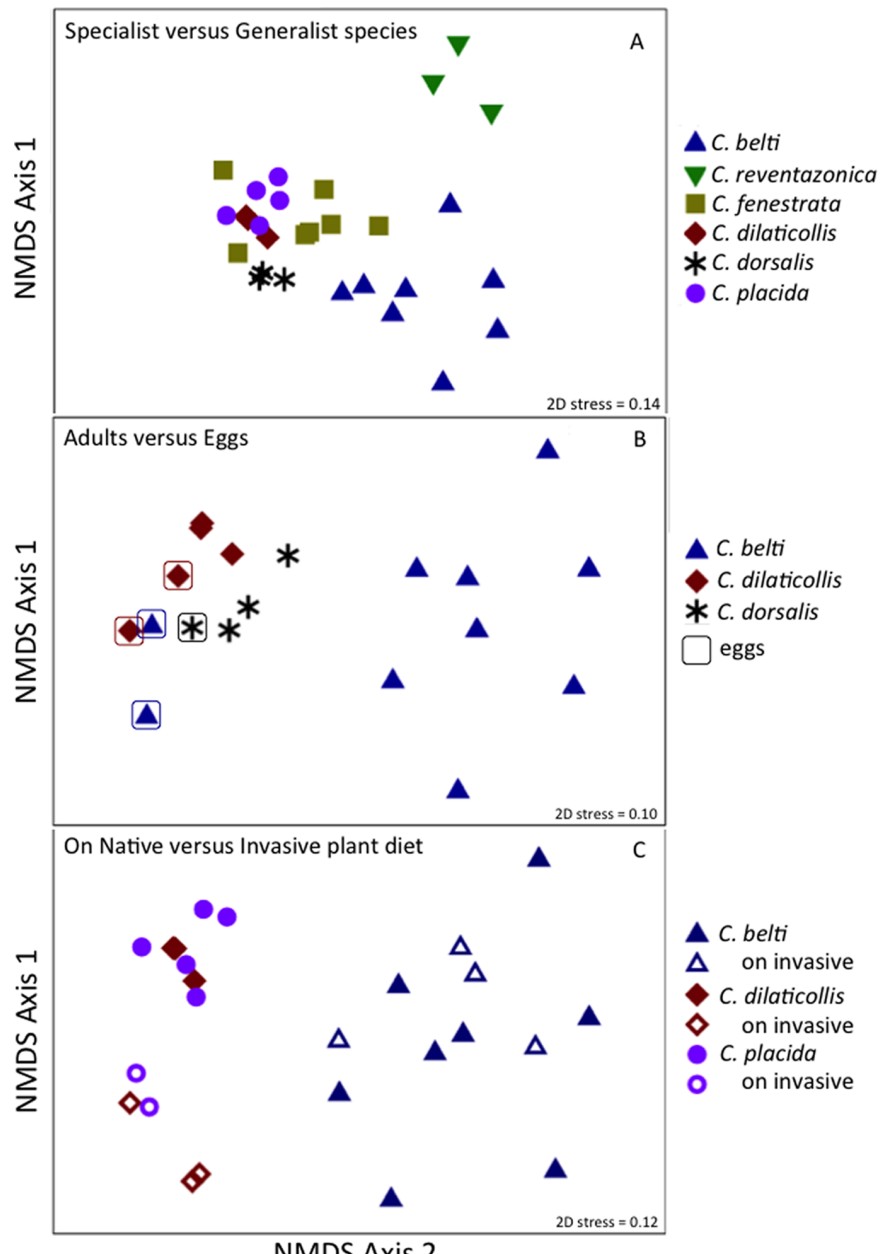

**Figure 6 Non-metric multidimensional scaling (NMDS) ordination of microbial communities associated with *Cephaloleia* beetles.** Each point represents all 16S rRNA sequences recovered from a single specimen. Displayed data was square root transformed, which minimizes errors in the ordination due to PCR bias while also not sacrificing genuine differences between samples. Samples with similar microbial communities plot closer together. ANOSIM $p$ values are shown. Lower stress values indicate better representation of the intersample (dis)similarities in two dimensions. (A) Ordination comparing four specialist species and two generalist species, the latter designated by triangles. $p = 0.001$, suggesting a distinct difference between the two feeding strategies. (B) Ordination showing the eggs of three species (outlined), compared to adults. $p = 0.060$ for all samples, suggesting a similarity to the communities found in specialist adults. $p = 0.022$ for the generalist species *C. belti* eggs versus adults, suggesting a significant difference between the two. (C) Ordination comparing three species, found on both native (filled symbols) and invasive plant species (open symbols). $p = 0.001$ for the two specialist species combined; $p = 0.05$ for *C. belti*, suggesting a significant difference in both cases.

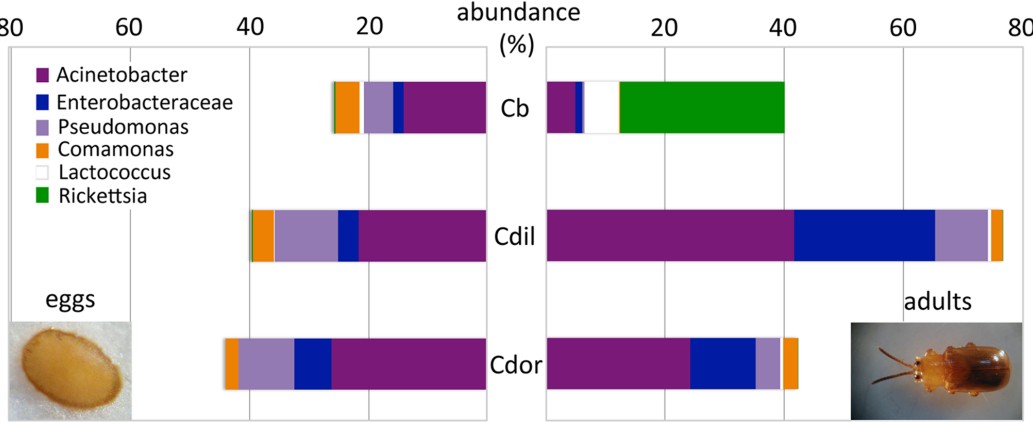

**Figure 7 Dominant bacterial 16S rRNA sequences, recovered from eggs and adults of *Cephaloleia* species.** Relative abundance of the six most dominant bacterial genera, based on 16S rRNA sequences, recovered from eggs (left) and adults (right) of three *Cephaloleia* species. Species abbreviations are as follows (*n* egg, *n* adult, respectively): Cb = *C. belti* (2, 8), Cdil = *C. dilaticollis* (2, 3), Cdor = *C. dorsalis* (1, 3). Photo credits: S. Goffredi.

to those on native plants (2%; $p = 0.012$; Fig. 2). The four specimens of the generalist species *C. belti* collected on pink banana (*Musa velutina*) also showed a slight shift in microbiome (ANOSIM $R = 0.27$, $p = 0.05$; Fig. 6B; Table S1), with significant increases in Spiroplasmataceae and Enterobacteriaceae abundance (ANOVA $p = 0.05$ for both; Fig. 2).

## DISCUSSION

Virtually every living organism has an associated collection of bacteria and bacteria-sourced genes (i.e., microbiome), which account for more genetic and functional potential than even the host genome. Insects have emerged to be important for this research due to their variable nutritional strategies and ecological dominance. The speciose genus *Cephaloleia* has evolved in association with tropical ginger plants and, for many species at La Selva Biological Station in northeastern Costa Rica, their specific host plant associations are known. Several *Cephaloleia* species are also currently expanding their diets to include exotic invasive plants, yet it is not known whether their microbial community plays a role in this transition. For this reason, Costa Rican rolled-leaf beetles within the genus *Cephaloleia* present a unique opportunity to distinguish the effects of host diet from host taxonomy on the associated gut bacteria, as well as to explore whether movement of these specialized insects onto invasive host plants results in changes to the bacterial communities. The factors that affect insect gut bacterial communities are still not fully understood. In particular, diet has been shown to affect gut microbial communities in some insects (*Colman, Toolson & Takacs-Vesbach, 2012*) and conversely be a poor predictor of gut bacterial community composition in others (*Jones, Sanchez & Fierer, 2013*). In this study, we show that the core microbiome of six closely related *Cephaloleia* species primarily includes the Moraxellaceae, Enterobacteriaceae, and Pseudomonadaceae, and that diet breadth is significantly linked to microbiome diversity and community structure.

Results suggest that the core recovered bacterial OTUs may be beneficial to *Cephaloleia* beetle hosts. For example, four of the most common bacterial OTUs associated with adult beetles were observed in eggs, suggesting likely pseudovertical transmission from mother to offspring. Sampling occurred over the course of 13 months, thus showing that these microbiome members likely have a non-transient relationship with their host. Additionally, *Pseudomonas*, *Enterobacter*, and *Pantoea* (also Enterobacteriaceae) have been found to play influential roles in development, nutrition, and success in other herbivorous beetles and true bugs (*Bistolas et al., 2014*; *Wang et al., 2016*; among others). A study by Minard et al. suggests that the mosquito *Aedes albopictus* specifically associates with *Acinetobacter* to help with digestion of plant nectar (*Minard et al., 2013*), while *Acinetobacter* and *Pseudomonas* in bark beetle digestive systems contribute to the nutritional requirements of the insect via the breakdown of plant-based compounds (*Briones-Roblero et al., 2017*; *Ceja-Navarro et al., 2015*). Representative isolates within the Enterobacteriaceae and Moraxellaceae revealed distinct in vitro metabolic capabilities with regard to the break down of plant-related compounds (e.g., xylose, mannitol, and pectin) versus lipids, respectively. Previously, Enterobacteriaceae were also found in *Bombyx mori* larvae (Lepidoptera) to similarly utilize mannitol and pectin, suggesting a role in the digestion of the mulberry leaf diet of the host (*Anand et al., 2010*). Some Chrysomelid beetles can degrade pectin and cellulose on their own, without the need of symbionts (*Pauchet, Wilkinson & Chauhan, 2010*; *Busch et al., 2017*), however, others lack this ability and, instead, possesses a highly specialized symbiont that degrades pectin (*Salem et al., 2017*). For the four species in this study, either situation may be possible. Additionally, many plants produce chemical toxins to defend against herbivory, and it is expected, although not well demonstrated, that bacterial symbionts could aid in detoxification of these defensive compounds (*Ceja-Navarro et al., 2015*). The primary food source of *Cephaloleia* beetles, the plant genus *Heliconia*, is known to produce polyphenolic compounds known as tannins (*Gibbs, 1974*). Interestingly, esterases in insects have also been shown to detoxify defensive plant compounds (*Snyder et al., 1998*), thus esterase-producing bacteria like *Acinetobacter* and *Pseudomonas* (*Salem et al., 2017*) could also provide this service to *Cephaloleia*. Preliminary experiments suggest that *Acinetobacter* isolates grow better in the presence of *Calathea* (Zingiberales) extract as the only source of nutrients (C. Blankenchip, 2017, unpublished). Tolerance to, and metabolism of, plant extracts hints at a possible role of either detoxification or direct nutrient acquisition by the dominant *Cephaloleia* microbiome, although further research with cultured bacterial isolates is necessary to examine this more fully.

*Cephaloleia* beetles in this study exhibited contrasting dietary breadths; the two generalist species *C. belti* and *C. reventazonica* feed on nine to 15 plants from many Zingiberales families, while the four specialist species, including *C. dilaticollis*, *C. dorsalis*, *C. fenestrata*, and *C. placida*, each feed on only one to two plant species (*McKenna & Farrell, 2005*; *García-Robledo et al., 2013*). This comparison of six congeneric *Cephaloleia* species with varying diet breadth satisfies the recommendation of *Jones, Sanchez & Fierer (2013)* to essentially remove taxonomy as a factor confounding the influence of diet on the gut microbiome. Overall, the microbiome of specialists was significantly higher

in diversity than generalists, and was comparatively dominated by the Moraxellaceae, Enterobacteriaceae, and Pseudomonadaceae. The shift shown in Fig. 6A perhaps suggests that part of axis 1 could be due to intrinsic species differences (e.g., *C. belti* and *C. reventazonica*, the generalist species, appear separated), but that axis 2 is likely driven by diet breadth (all specialists cluster tightly together). It is worth noting here that *C. dilaticollis* encompasses two cryptic species, with opposing diet breadths, and that the individuals in this study possessed a microbiome community shared with other specialists. Thus, it was possible to infer the limited diet breath of this particular *C. dilaticollis* sub-species, based solely on a distinctive microbiome structure and diversity. The specialist sub-species of *C. dilaticollis* was subsequently confirmed via insect COI sequencing, in consultation with Dr. Carlos Garcia-Robledo (University of Connecticut).

Generalist *Cephaloleia* beetles were, by contrast, colonized by bacteria traditionally thought of as pathogens, including *Rickettsia* and *Rickettsiella*, with a pattern of occurrence in these beetles (Fig. 3, shown in green) consistent with a pathogen-like relationship (*Azad & Beard, 1998*). They infect only a few individuals (and thus exhibit low prevalence), but when present, they achieve high numbers (and thus high abundance). Individuals that were colonized by these groups demonstrated a striking paucity of several of the most prevalent "core" microbiome members observed in all other beetles, including *Acinetobacter* and Enterobacteriaceae (Figs. 2 and 5). *Sakurai et al. (2005)* similarly showed that *Rickettsia* presence in aphids reduced the population of the beneficial bacterial symbiont *Buchnera* to 50–60% of its density in *Rickettsia*-free individuals. If these interloper microbial groups are detrimental, it would follow that beetles demonstrating dysbiosis would suffer fitness deficits, including weight loss and poor survival. Indirect observations support this assertion, in that the smallest *C. belti* individuals appeared to have a higher incidence of *Rickettsia* colonization (75%, as compared with 37% for the largest individuals, $n = 8$ in both groups). In other studies, *Rickettsia* has had a positive effect on insects, including higher fecundity, faster development, and fungal resistance (*Sakurai et al., 2005*; *Himler et al., 2011*; *Łukasik et al., 2013*). Whether beneficial or pathogenic, it will be interesting to further examine the possible antagonistic relationships among members of the *Cephaloleia* microbiome.

Over the past several years, at least eight *Cephaloleia* species at La Selva Biological Station have been found foraging on invasive crêpe ginger, false bird-of-paradise, pink velvet banana, and white ginger (*Schneider, Rasband & Eliceiri, 2012*). In this study, specialist *Cephaloleia* species collected on invasive plants exhibited an apparent dysbiosis in the membership of both core groups, the Moraxellaceae and Enterobacteriaceae, and non-core groups Brevinemataceae and Spiroplasmataceae It is not known if these differences represent a change along a continuum as beetles adapt to exotic plants, or whether the changes in microbiome facilitate movement onto new plants, or neither. Determining whether an elastic microbial repertoire can be a form of direct, and rapid, environmental adaptation by the host is a next critical step given that the colonization of invasive plants is an inevitable new reality for all generalist and specialist herbivores. A 2013 NSF-sponsored report urged the scientific community to better understand phenotypic plasticity and sensitivity of animals to future changing environments

(*Padilla et al., 2014*), yet none of the statements considered animal-associated microbiomes, or the immense potential of this metabolic reservoir for maintaining function in the face of changing ecosystems.

## CONCLUSION

The tremendous diversity of insect herbivores, particularly in tropical rainforests, is due in part to the relative specificity of their diets (*Novotny et al., 2006*; *Forister et al., 2015*). In this study, Costa Rican beetle species within the genus *Cephaloleia*, with known diet breadths ranging from generalist (foraging on over nine plants) to specialist (foraging on less than two plant species), were analyzed for their associated gut microbial community. The core microbiome of six closely related species of Costa Rican *Cephaloleia* beetles was limited and mainly included members of the *Acinetobacter*, Enterobacteriacea, *Pseudomonas*, *Lactococcus*, and *Comamonas*. Contrary to expectations, the microbiome diversity was significantly higher in specialist species, compared to generalists, and was dominated by these core groups (as were the eggs). Generalist beetles had lower diversity, primarily due to the exclusive dominance of bacteria thought to be pathogens, including the Rickettsiaceae. Bacteria isolated from *Cephaloleia* digestive systems had distinct capabilities in both digestion of plant-based compounds, including xylose, mannitol, and pectin, and possible detoxification of plant compounds via lipases. Additionally, changes in abundance of rare plants may significantly influence the balance between nutritional specificity and dietary breadth of herbivores (*Norton & Didham, 2007*), and in this study, *Cephaloleia* specimens collected from exotic invasive plants revealed a dysbiosis of the microbiome. In ecological terms, it may be possible in the future to use only microbiome patterns to know whether the beetles are surviving in native or exotic environments. Additional experiments are necessary to fully determine whether microbiome differences observed in this study are the product of intrinsic differences among species or result from shifts to novel plant diets, and what are the implications of replacement of native plants by exotic plants on the survival of beetles. The possible relationship between gut bacteria and niche adaptation, however, remains an important and urgent research question as organisms respond to future altered landscapes.

## ACKNOWLEDGEMENTS

The authors thank Dr. Erin Brinton and Natalie Gonzalez for assisting in sample collection while in Costa Rica, Dr. Gretchen North for providing the intellectual support to identify plants, Dr. Carlos Garcia-Robledo for sharing his knowledge of *Cephaloleia*, the staff and administration of La Selva Biological Station, and the members of the Occidental College Microbial Symbiosis Laboratory.

### Funding

Funding for this project is provided by the Fletcher Jones Science Scholars Award, the Occidental College Undergraduate Research Center Academic Student Project Grant, and

the Occidental College International Programs Office. The funders had no role in study design, data collection and analysis, decision to publish, or preparation of the manuscript.

### Grant Disclosures
The following grant information was disclosed by the authors:
Fletcher Jones Science Scholars Award.
Occidental College Undergraduate Research Center Academic Student Project Grant.
Occidental College International Programs Office.

### Competing Interests
The authors declare that they have no competing interests.

### Author Contributions
- Chelsea L. Blankenchip conceived and designed the experiments, performed the experiments, analyzed the data, prepared figures and/or tables, authored or reviewed drafts of the paper, approved the final draft.
- Dana E. Michels performed the experiments, analyzed the data, approved the final draft, collected samples in Costa Rica.
- H. Elizabeth Braker conceived and designed the experiments, contributed reagents/materials/analysis tools, approved the final draft, contributed insect ecology expertise.
- Shana K. Goffredi conceived and designed the experiments, analyzed the data, contributed reagents/materials/analysis tools, prepared figures and/or tables, authored or reviewed drafts of the paper, approved the final draft.

### Field Study Permissions
The following information was supplied relating to field study approvals (i.e., approving body and any reference numbers):

Field experiments were approved by the Costa Rican Ministry of the Environment and Energy. Permit #R-026-2015-OT-CONAGEBIO.

### DNA Deposition
The following information was supplied regarding the deposition of DNA sequences:

16S rRNA sequences for bacterial isolates are available from GenBank under accession numbers MF776885–MF776899.

### Data Availability
The raw barcode sequence data and QIIME processed data are available from the Dryad Digital Repository: http://dx.doi.org/10.5061/dryad.5fj6t. Raw sequences were aligned and quality control for unidentified base pairs and chimeras was performed according to the specifics noted at https://sites.oxy.edu/sgoffredi/Symbiosis_Lab/LabScripts.html.

### Supplemental Information
Supplemental information for this article can be found online at http://dx.doi.org/10.7717/peerj.4793#supplemental-information.

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
