# Peer review of "Diet breadth and exploitation of exotic plants shift the core microbiome of Cephaloleia, a group of tropical herbivorous beetles"

_PeerJ, doi:10.7717/peerj.4793_

## Round 0.1 · original submission · Minor Revisions

Please address each reviewer comment in the manuscript (indicating which line numbers) or provide a response as to why the comment was not incorporated.

Reviewer 1 ·

Basic reporting

See comments in: General comments for the author

Experimental design

See comments in: General comments for the author

Validity of the findings

See comments in: General comments for the author

Additional comments

The work of Blankenchip et al. “Diet breadth and exploitation of exotic plants shift the core microbiome of tropical herbivorous beetles“ makes an important contribution because it aims to “to characterize the diversity of bacteria, associated with adults and eggs in species of Cephaloleia beetles: generalist and specialist species”.
Changes in abundance of rare plants may significantly influence the balance between nutritional specificity and dietary breadth of herbivores and may therefore affect the biodiversity of forest-dependent animal species.
Neotropical forests are extremely diverse ecosystems, listed within the biological hotspots of our planet. However, species from these forests are threatened by diverse anthropogenic activities: among them, the replacement of native vegetation by exotic plantations (monocultures), which justifies the accomplishment of this work, although this aspect was little approached in the study of Blankenchip and colleagues.
In general, the Manuscript at hand is well written. Data and analysis performed by authors seems to be the appropriate for this kind of studies to explore the study proposal. The figures and tables are illustrative, well addressed in the text and the captions are adequate and informative.
However, I believe the work needs some improvements, as specified below:

1. Title: This study refers to a specific group of beetles and not to all herbivores. I believe the title should refer to it.

2. Introduction: Why did the authors choose to study only this genus, since there are several other species of herbivores? It should be explained in the introduction.

3. Introduction (lines 67-68): Scientific names, during the first presentation, should have full name + author.

4. Discussion (L 313-324): I think this information gets more interesting in the introduction (see my comment nr 2).

5. Discussion (L 330-331) Which lines?? Mention some literatures that show these lines ..

6. Discussion / Final remarks: (Suggestions to include in the work):
a) What is the implication of the replacement of native plants by exotic plants in the survival of beetles (or other species)? Is it possible to reach these conclusions with your results?
b) In ecological terms, is it possible to use only bacteria to know if the beetles are surviving in native or exotic environments?
c) What are the challenges for the conservation of species of beetles or other groups with the implantation of exotic plants?

7. References: Standardize the formatting of references. Some are complete and others are abbreviated.

Reviewer 2 ·

Basic reporting

No comment

Experimental design

No comment

Validity of the findings

No comment

Additional comments

In the manuscript entitled “Diet breadth and exploitation of exotic plants shift the core microbiome of tropical herbivorous beetles” Blankenchip and colleagues describe the microbiome of different beetle species from the Cephaloleia genus. The article presents a very interesting experimental set up where they study the microbiome of 6 different beetles belonging to the same genus, but with different diet breadths. Whereas 4 are specialists, two are generalists.
The authors conclude that all species share a core microbiome consisting of 8 OTUs, and that diversity and community structure is linked to diet breadth, with specialists having higher diversity. They also observe that the microbiome of some beetles feeding on native plants is different than those of the same beetles feeding on invasive plants, and suggest that the latter is a dysbiotic state that could be involved in niche adaptation.

I find this body of work extremely interesting, I really liked the experimental set up and the paper is very well written. The questions asked are central to the field of symbioses. I just have some minor comments that are easily fixable.

First, Blankenchip and colleagues found Wolbachia in the microbiome of all species, but was removed from further analysis arguing “its known prevalence in insects as a reproductive pathogen”. I believe this may not be a good reason, given that there are instances in the literature where Wolbachia seems to be found in the gut of insects, and also examples where they are not reproductive pathogens, but rather beneficial partners. Since most of the discussion seems focused on gut microbes, are the authors sure that Wolbachia is not located in the gut?

Second, the authors claim that the presence of some OTUs both in the adult beetles as well as on the eggs is suggestive of vertical transmission. However, this is not necessarily the case. The authors do not give enough detail on how the eggs were collected (please include this in the materials and methods section), but it seems that they were obtained from bags were adults had been for a while. The presence of these OTUs on the egg surface may be just contamination after being in the same bag as the adults, where presumably they could have defecated, for instance. If the authors do not want to perform experiments to test the mode of transmission, they should at least mention the possibility of contamination.

Third, the authors suggest the role of the microbiota to be beneficial for the beetles, and put forward various ideas such as degradation of plant complex polymers and secondary metabolites. Cephaloleia beetles belong to the Chrysomelid family. Kirsch and Pauchet have produced a body of work on the degradation of plant polymers in Chrysomelids. They have found that often beetles can degrade pectin and cellulose on their own, without the need of symbionts. Salem et al. 2018 showed that one Chrysomelid species lacks this ability on their own genome, and possesses a highly specialized symbiont that degrades pectin. Either of these cases may, of course, not be the way things work for Cephaloilea, but I feel this literature should be mentioned if the hypothesis is going to be put forward.

Also, the authors suggest detoxification as a possible role for the microbiome, but never mention why these beetles should detoxify. What are the toxic compounds present on the leaves (alkaloids, and polyphenols)? Are these compounds known to be degraded by bacteria? A good reference here would be the work by Ceja Navarro and colleagues showing caffein being degraded by Pseudomonas in the coffee bean borer which leads to fitness benefits for the host beetle.

Minor comments:

Please, check the figure numbers. They should appear in the text in order, Figure 1, Figure 2…etc. Currently, Figure 2 is at the end of the results section. Figure 7 and 6 seem also not to be in order.

L62: the word “exclusively” sounds weird, since the beetles use the plants for almost everything: nutrition, development, reproduction and shelter.

L98: what do the authors mean by “partially dissected beetles”?

L147: how was the representative sequence for each OTU selected: random, longest, most abundant, first picked…(parameter -m in Qiime)?

L248: are these values Shannon indexes, number of OTUs?

L279: where these OTUs that are shared between eggs and adults the exact same OTU (same OTU number), or just same taxonomical binning?

L329: Briones-Roblero and colleagues (reference 34) only show the ability of some members of the microbiota to degrade plant compounds and only put forward the hypothesis that this may be beneficial for the insect. A better example would be that of Ceja Navarro mentioned above.

L332: I am not sure the word “complementary” is the correct one here. It somehow implies that one cannot work without the other because they complement each other. Rather, they are distinct functions of the microbiome, but independent from each other.

Figure 2 and 5: please use the same species name format as in the other figures (C. belti, C. reventazonica etc.)

Figure 4: what does grey and 0.5 mean? Please, explain.

L397: authors observe that microbiome diversity contradicts their previous hypothesis that it should be higher in generalists than specialists. This is a very cool result, but it would be nice to elaborate on why this could be.

L500: Typo in “Degradation”

---

## Round 0.2 · accepted · Accept

Thank you for carefully addressing the reviewer comments.

#